# Cost-Effective Multiplex Fluorescence Detection System for PCR Chip [note 1]

**DOI:** 10.3390/s21216945

**Published:** 2021-10-20

**Authors:** Sung-Hun Yun, Ji-Sung Park, Seul-Bit-Na Koo, Chan-Young Park, Yu-Seop Kim, Jong-Dae Kim

**Affiliations:** 1School of Software, Hallym University, Chuncheon-si 24252, Korea; butter9709@gmail.com (S.-H.Y.); rntmfqlcsk@gmail.com (S.-B.-N.K.); cypark@hallym.ac.kr (C.-Y.P.); yskim@hallym.ac.kr (Y.-S.K.); 2Bio-IT Research Center, Hallym University, Chuncheon-si 24252, Korea; 3Biomedux Co., Ltd., Suwon-si 16226, Korea; jspark@biomedux.com

**Keywords:** real-time PCR, multiplex, filter wheel, PCR chip, point of care, open platform camera

## Abstract

The lack of portability and high cost of multiplex real-time PCR systems limits the device to be used in POC. To overcome this issue, this paper proposes a compact and cost-effective fluorescence detection system that can be integrated to a multiplex real-time PCR equipment. An open platform camera with embedded lens was used instead of photodiodes or an industrial camera. A compact filter wheel using a sliding tape is integrated, and the excitation LEDs are fixed at a 45° angle near the PCR chip, eliminating the need of additional filter wheels. The results show precise positioning of the filter wheel with an error less than 20 μm. Fluorescence detection results using a reference dye and standard DNA amplification showed comparable performance to that of the photodiode system.

## 1. Introduction

Infectious diseases continue to impose a major burden to global health and the economy [1,2,3,4]. To prevent an epidemic or pandemic outbreak, highly contagious diseases such as influenza require constant monitoring for early diagnosis. Especially in the case of influenza, a sensitive and rapid diagnostic device to detect and analyze the virus is essential in preventing further spread [5]. In addition, research shows that prompt detection and analysis of influenza for children with fever at emergency rooms results in the reduction of time, cost, and need of additional diagnosis [6,7,8,9].

Emerging technologies that increases the efficiency and reduces the time of diagnosis has led to the development of various rapid diagnostic platforms which can be employed at point-of-care (POC) [10,11]. By applying POC devices to diagnose just four common diseases, including bacterial pneumonia, syphilis, malaria, and tuberculosis can prevent 1.2 million deaths annually [12,13].

Despite extensive efforts and improvement in global health, preventing death caused by infectious diseases in developing countries remains a challenge [14]. Viruses such as Ebola, Zika, Chikungunya, Dengue, Malaria, HIV, and other emerging pathogens have a high probability of causing a global pandemic owing to their high infectivity, and taking an especially hard toll on the economy and health of developing countries where resources are scarce. Deploying POC diagnostic devices to these settings can address these challenges by improving the currently limited medical system and enhance the medical monitoring and maintenance status.

Three major diagnostics methods used at POC are viral culture, serological diagnosis, and nucleic acid detection. Nucleic acid detection is favored over the other two methods owing to its simplicity that does not require highly skilled professional and expensive equipment as in viral culture, and eliminates the complex step of antibody synthesis that serological diagnosis requires [1,15,16]. In particular, polymerase chain reaction (PCR) that replicates nucleic acids to billion-fold is the gold-standard of pathogenic marker detection [17,18]. Since PCR detects the unique nucleic acid sequence of each pathogen in the ribonucleic acid (RNA) or deoxyribonucleic acid (DNA), it provides high specificity and accuracy. The recent COVID-19 pandemic outbreak highlights the advantages of PCR, with the real-time PCR (qPCR) becoming the official diagnosis method for COVID-19 [19,20]. 

Although PCR is a simple and straightforward diagnosis method, the operation is mainly carried out in certain facilities such as in hospitals or central laboratories due to portability and cost limitations, which can be disadvantageous in emergencies [1,21,22,23,24,25,26]. For example, there are cases where administration of Tamiflu within 48 h of initial symptom observation is critical for influenza patients [27]. Rapid, on-site diagnosis of pathogens for patients admitted to emergency rooms suspecting respiratory diseases, gram-negative bacteria, and tuberculosis is crucial to determine the right antibiotics and treatment [1,28]. 

According to the World Health Organization (WHO), POC tests that are ideal for healthcare in resource-limited settings should meet the criteria of ‘ASSURED’, which stands for affordable, sensitive, specific, user-friendly, rapid and robust, equipment-free, and deliverable [29,30]. Recent studies developing POC platforms aim to construct a device that is portable, low cost, delivers rapid results with less sample volume and has a user-friendly interface to meet the criteria [1,27,31,32].

Commercially available qPCR equipment shows high accuracy and sensitivity in detection. However, it is difficult to meet the ‘ASSURED’ criteria and use the equipment at resource-limited settings since it incorporates highly a sensitive fluorescence detection method, increasing the cost to 10-fold that of a conventional PCR platform [33,34,35,36,37]. To overcome these limitations, various microfluidic chips and low-cost fluorescence detection methods have been reported [23,26,38,39,40,41]. El-Tholoth et al. (2021) used real-time reverse transcriptase loop-mediated isothermal amplification (qRT-LAMP) to lower the thermal cycle cost, but the cost of fluorescence detection remained an issue because they used a universal serial bus (USB) fluorescence microscope. An et al. (2020) also proposed a microchip capable of low-cost thermal cycling, but used a relatively expensive digital camera for fluorescence detection. When a digital camera or an industrial camera is used, not only the cost of the camera itself but also that of the emission filter becomes a problem because the filter size becomes larger in proportion to the lens diameter. Note that the cost of the interference filter, which is mainly used as a fluorescent filter, increases rapidly depending on the size [42]. Mendoza-Gallegos et al. (2018) implemented a low-cost thermal cycler using a power resistor and a fan with a Raspberry Camera Module V2 to lower the detection cost, but rearranged the camera lens and filter to place the emission filter directly above the image sensor. This led to the cost increase for optic assembly and difficulty in designing the filter wheel for multiplex fluorescence detection.

Smartphone cameras are also used as a major approach to lower the cost of fluorescence detection [14,39,41,43]. However, due to the rapid development of smartphones, camera modules for smartphones equipped with standard interfaces such as mobile industry processor interface (MIPI) and USB are continuously being released at low prices. Therefore, the smartphone camera module can replace smartphone cameras itself without significant cost burden. In addition, since the smartphone camera module is usually equipped with an autofocus lens with a small diameter, the size of the emission filter can be reduced resulting in a significant reduction in cost. Since all the aforementioned studies perform single target detection, a low-cost compact emission filter wheel, such as that proposed in this paper, is required to extend to multiplex detection.

The system proposed in this paper allows for low-cost thermal cycling using a previously reported PCR chip based on a printed circuit board (PCB) with attached thermistor and heater patterns [44,45]. A high-performance and low-cost smartphone camera module and small size emission filters were adopted for fluorescence detection. Due to the small size of the emission filters, the overall size of the filter wheel could also be reduced. In addition, the small filter wheel size allows the use of small linear stepper motors. The excitation module with four LEDs is constructed with a side-illumination method to hold the light source at an angle, eliminating the need for an additional filter wheel. 

The precision validation experiment proved that the proposed multiplex fluorescence detection mechanism can reliably detect the target fluorescence. To evaluate the fluorescence detection performance, four standard fluorescence dyes were selected and tested individually, and as a mixture to investigate the cross interference between the dyes. Finally, qPCR quantification during actual reaction was validated by amplifying and detecting a standard *Chlamydia trachomatis* DNA. Together, these results demonstrate that the proposed multiplex qPCR is suitable as POC test equipment.

## 2. Materials and Methods

### 2.1. Overall System

The PCR chip previously reported by our group consists of a PCB and a reaction chamber, where the reaction chamber is in the shape of a water drop and is flat as seen in Figure 1a as opposed to the conventional tube format. The chamber was made with polycarbonate using a mold, enabling fluorescence detection at the top owing to the transparency of the material, in which the spatial distribution of the fluorescence can also be observed. The reaction chamber is in contact with the heater pattern on the black matte PCB using a 100 μm medical grade double sided tape (1510, 3M, Saint Paul, MN, USA), and the thermistor is attached at the back of the PCB. The assembled PCR chip with the exterior housing is shown in Figure 1b. This simple structure allows rapid thermal cycling with less sample volume, and further simplifies the optics of the detection system. 

Figure 2 illustrates the functional block diagram (Figure 2a) and schematic of the proposed system (Figure 2b), consisting of the PCR chip, excitation unit, filter wheel, and open platform CMOS camera. The PCR chip and excitation unit is controlled by the microcontroller system that was previously reported by our group [44,45]. The USB video class compliant camera made with Sony IMX179 image sensor (HBVCAM-8M1822 V22, Huiber Vision Technology Co., Ltd., Shenzhen, China) was selected. The IMX179 has been adapted various smartphones and has a cell size of 1.4 µm × 1.4 µm, the maximum resolution of 3264 × N2448, and the diagonal size of 5.7 mm. The camera is embedded with an autofocus lens and provides automatic white balance, automatic gain, and automatic focusing. The focus was adjusted manually because autofocusing did not work well in a dark environment of fluorescence photography. Moreover, since the purpose was to measure the fluorescence quantity, the automatic gain was turned off and the gain was set to the lowest value. No additional lenses, such as collimation lenses, were used. Since the field of view (FOV) is 70°, the emission filter requires a size that is 1.4 times larger than the distance of the camera to the end of the emission filter to prevent blind sights. The motion rail for the filter wheel of the emission unit was made with aluminum and sliding tape, further increasing the compactness and cost-effectiveness. The whole system is concealed with a cover during experiments to prevent the influence of ambient light.

### 2.2. Excitation Unit and Emission Filter Wheel

The mechanism of the proposed system was determined using fluorescein (FAM), hexachloro-6-carboxyfluorescein (HEX), 6-carboxyl-X-rhodamine (ROX), and cyanine 5 (CY5). Therefore, the excitation unit includes light emitting diodes (LED) and excitation filters for the selected four fluorescence dyes. If a light bulb or a white LED is used for excitation, an additional excitation filter wheel is required for multiplex detection. Even if the excitation filter wheel and emission filter wheel is combined, the cost effectiveness, size, and simplicity of manufacture is restricted. In the proposed system, four appropriate LEDs to excite the fluorescence dyes are fixed near the PCR chip so that the incidence angles to the chip surface of their light passing through the excitation filters are 45 degrees, as shown in Figure 3b. 

The cut-off wavelengths of the excitation filter and the emission filter are shown in Table 1, and the average optical density (OD) of all filters are ≥6.0. Table 2 shows the specifications of the employed LEDs. A radial type LED with the diameter of 5 mm was used for excitation, and the diameter of the camera lens was 3 mm. To meet the dimension specifications of the camera lens and excitation LEDs, all the filters have the diameter of 5 mm and thickness of 1 mm (Chroma Technology Corporation, Bellows Falls, VT, USA).

To implement the filter wheel for the emission unit, the components shown in Figure 4a were made using aluminum with the thickness of 1 mm, and then assembled on an aluminum base with the same thickness. The holder for the filters has five holes in total, where four are for the emission filters and an additional 3 mm hole is at the far-right side to hold a 3 × 1 mm magnet used to retrieve to home position. The emission filter and magnet were secured in place using an optically transparent tape. To maximize light shielding, a 5 mm hole was also drilled in the aluminum base to block lights that may come from the sides, where the camera will be positioned. Note that the filtering effect of obliquely incoming light is less effective because the emission filters are interference type. Therefore, the emission filter needs to be able to align with the aluminum base hole to ensure clear FOV during each cycle. A sliding tape was applied at the guide rail edge of the holder and the inside of the holder cover to ensure smooth mobility (ASF-110FR, Chukoh Chemical Industries, Ltd., Tokyo, Japan). The finger of the holder, which is also made with a 1 mm thick aluminum, was attached to the filter holder using a strong adhesive (Loctite 401, Henkel Ltd., Hemel Hempstead, UK). The holder finger has grooves with a width that matches the carriage of the linear stepper motor. The linear stepper has operated with a stepper motor with the diameter of 10 mm, and has the horizontal stroke of 34 mm (Micro 5V 2-phase 4-wire Stepper Motor Precision Linear Actuator Screw Slider Nut, Micro Motor, China), which is operated with an Arduino microcontroller (Qt Py, Adafruit, New York, NY, USA) and a motor driver (L293DD, STMicroelectronics, Geneve, Switzerland). The microcontroller controls the positioning of the filter holder, and secures the home positioning after each cycle using the motor home position sensor interface. 

Fluorescence detection starts at the end of each annealing step of the PCR cycle, acquiring images of all four fluorescence in sequential order starting from the emission filter that is closest to the camera when set at home position. The filter holder is prepositioned at the home position before acquisition starts and returns after the images for all four fluorescence are acquired. Figure 5 shows the filter wheel operation when the first filter to be visited is at the second filter position from the magnet. Figure 5a shows the schematic of the filter wheel. and Figure 5c shows that the filter holder is always positioned at home position before the reaction starts, and after all four images acquired per cycle. For rapid home positioning, coarse/fine two step searching method is employed using a magnet and a hall sensor. The filter holders rapidly move towards the right until the hall sensor noticed the magnet as shown in Figure 5b. During this coarse search step, the position may not be accurate due to the high speed of movement. After the coarse search, the holder starts slowly moving to the left until the polarity of the hall sensor signal is inverted (fine search) allowing precise positioning. Figure 5d represents a setup when the first desired fluorescence image requires the emission filter located on the second position of the filter holder. Meanwhile, the camera exposure time was set to 0.5 s so that the fluorescence image for the maximum DNA concentration had maximum intensity but was not saturated. Since open platform cameras usually do not have a strobe signal to synchronize the lighting, the LED excitation time was set to 3 times the exposure time to obtain a fully exposed image frame. The resultant time for taking a photo was 1.5 s. It takes 9.1 s to visit all emission filters and save the four-channel image from the camera, which is 12% compared to the time per PCR cycle (75 s).

Errors on the filter holder position will result in a misalignment of the base hole and the emission filter, narrowing the FOV. A test fixture in which the entity of the filter wheel can be seen through the camera was constructed to evaluate the motor precision. The fixture was made as a dark room with a white LED strip illumination to prevent influence from outside light sources. The cover and filter holder were removed from the filter wheel, and a blue circular sticker with the diameter of 9 mm was aligned with the base hole for image acquisition and analysis to determine spatial resolution around the base hole. Converting the image acquired from the camera to a HSV space shows clear distinction of the blue sticker compared to the background (Figure 6a). From this converted image, a bounding box was obtained that only contains the blue sticker, having 400 pixels through the diameter resulting in 20.4 µm per pixel.

To investigate the filter positioning precision, the filter holder and cover was assembled to the filter wheel, and a white paper was placed at the bottom of the base to distinguish the holes on the holder. The emission filter was not attached to the filter holder for this experiment. The images acquired through camera should be close to a perfect circle when the filter holder position is precise. On the other hand, the images will show an ellipse when there is a positioning error due to the overlap of the filter and base holes. This causes the variation of the centroid of the hole image. Therefore, the position error can be evaluated by image binarization and calculating the object centroid. The filter position error in pixels can easily converted to in micrometer with the resolution obtained before. Sequential acquisition of each four filter holes and repositioning to home position prior to the next acquisition cycle was repeated 84 times to determine the precision of positioning.

### 2.3. Flourescence Detection Performance Analysis

To achieve accurate and consistent analysis of fluorescence during reaction, a process to crop just the chamber region from the captured fluorescence image is necessary. Since the PCR chip is physically fixed on the chip holder in front of the excitation unit and does not move throughout the reaction, the fluorescence can be analyzed with consistency using a calibration chip and obtaining the region of interest (ROI), i.e., the chamber.

The process of obtaining the ROI using calibration chip is shown in Figure 7. The calibration chip image was converted to a gray image and filtered before binarization, and the filtered gray image was binarized using the Otsu algorithm. Binary noise caused by halos around the chamber, reflection from the heater pattern or reagent entrance was eliminated by applying the 5 × 5 opening morphological filter thrice. The resultant binary image is shown in the middle of Figure 7, and the ROI was selected to be where the horizontal and vertical projection was over the threshold. The rightmost image in Figure 7 shows the image of just the ROI of the calibration chip obtained through the process. The reference fluorescence unit (RFU) is defined as the average of the intensity within the ROI. The fluorescence detection performance of the system was evaluated using the four aforementioned standard fluorescence dyes and tested individually, and as a mixture to investigate the cross interference between the reagents (crosstalk experiment). For both experiments, the concentration of 0.56 pmole/µL was used for all dyes, which represent the maximum saturation fluorescence for DNA. The individual tests compared RFU from the image of the PCR chip with each individual dye to that with double distilled water (DDW). For the crosstalk experiments, RFU of the chip with the mixture of all four dyes was compared to that of the chip where the desired fluorescence dye was excluded from the mixture. This will allow the selectivity of the fluorescence detection. Furthermore, if the fluorescence intensity acquired in the individual experiments and that acquired for a particular dye in the crosstalk experiments are similar in value, it can be stated that there is no interference of the other dyes during detection. Such comparison between the individual and crosstalk experiments will demonstrate the influence of interference of any reagents when two or more emit fluorescence within the sample.

To evaluate the qPCR quantification efficiency, *Chlamydia trachomatis* DNA was subject to PCR with the initial copy number of 10^6^ per reaction, in a total reaction volume of 36 µL (Table 3). According to the primer design given by the manufacturer (Biomedux, Suwon, Korea), the fluorescence for FAM was detected and analyzed. Table 4 shows the thermal cycling profile for the PCR reaction, where the fluorescence intensity was acquired at end of the annealing step.

The qPCR performance of the proposed system was compared to that from real-time PCR chip system (Xavier^TM^, Biotmedux, Suwon, Korea), which is a validated qPCR equipment that employs photodiode detectors as the fluorescence detection method.

## 3. Experimental Results

The emission filters for FAM, HEX, ROX, and CY5 were attached to the filter holder and the fluorescence images were taken as described in the motor precision experiment. The hole from each fluorescence image is cropped and shown in Figure 8a, where the white dots represent the centroid of hole in the binary image. Fluorescence detection was repeated 84 times for all four fluorescence dyes, and the centroid location in each experiment is plotted as a scatter plot in Figure 8b. The results show less than 1 pixel variation both vertically and horizontally, which corresponds to less than a 20.4 µm position error rate.

Figure 9 shows the results from the individual and crosstalk fluorescence detection experiment. The fluorescence detection for individual dyes compared to DDW is shown in Figure 9a in which ‘No dyes’ row shows the image for DDW and the ‘Target dye’ row shows the fluorescence image acquired for the corresponding column label fluorescence dye. For the crosstalk experiment, the fluorescence image acquired with all four mixed dyes are labeled as ‘All dyes’, whereas ‘Except target dye’ represents the image when the three fluorescence dyes except the one depicted in the column are mixed (Figure 9b). As can be seen, the fluorescence of the ‘Target dye only’ row in Figure 9a and ‘All dyes’ row of Figure 9b show similar fluorescence, confirming that there is negligible interference between dyes in this system.

For fluorescence quantification, the background fluorescence is always subtracted from the intensity of the target fluorescence. Since the image without the target dye represents the background fluorescence, this was subtracted from the fluorescence intensity of the target to compare the gap between the two RFUs. On the other hand, when the excitation becomes stronger, the background florescence and the gap increase together. Therefore, the gap needs to be normalized with the background florescence for accurate evaluation, where the normalized gap is referred to as relative gap. Both gaps and the relative gaps are summarized in Table 5. For the individual dye experiment, the gap and relative gap are calculated as the absolute difference between ‘Target’ and ‘No dyes’ and dividing the gap with ‘No dyes’, respectively. This was carried out similarly in the case of the crosstalk experiment. Although the gap increased slightly for the crosstalk experiment compared to the individual experiment, the relative gap was above 9.5 for all conditions, with less than ±2 difference for each dye between the individual and crosstalk experiments. In addition, the relative gap tends to decrease in the crosstalk experiment with the exception for ROX. This result indicates that the fluorescence interference between the dyes is marginal and can be neglected.

Given that the proposed system can successfully detect multiple fluorescence signals with high precision and accuracy, the qPCR quantification performance was evaluated and compared to that from a reference system (Figure 10). The fluorescence images of the ROI acquired during PCR of 40 cycles are shown in Figure 10a and rapid increase in intensity is observed after the 27th cycle. Since the proposed system has a different scale compared to the reference system, normalization of the acquired data is necessary. Normally in qPCR, the fluorescence signal from the first few cycles is used to determine the baseline fluorescence which can be interpreted as the background signal. To account for the difference the background across equipment, baseline correction is a crucial step in qPCR analysis and is generally integrated into the equipment [46]. Therefore, the fluorescence detected in the proposed system was corrected according to baseline determined by initial 10 cycles and scaled to have the same RFU of 40th cycle as the reference system. The result is plotted with that from the reference system (Figure 10b). The plot shows that the proposed system delivers comparable results to that of the reference system. 

Figure 11 shows the cycle threshold (C_q_) of the proposed and reference system, which is obtained by comparing the logarithm of the fluorescence value acquired to a predefined threshold. In other words, the C_q_ is determined as the cycle number where the logarithmic curve of the fluorescence intersects with the predefined threshold. The log threshold was set to be 5.47 as provided by the reference system, and C_q_ was calculated using linear interpolation [46]. The results show a difference of 0.3 for the C_q_ between the two systems, demonstrating that the performance of the proposed system is comparable to the reference system.

## 4. Discussion

This research proposes a compact and cost-effective multiplex fluorescence detection system utilizing the PCB-based PCR chip previously reported from our group [46]. The portability and cost-effectiveness of the multiplex fluorescence detection system was drastically increased by employing an open platform CMOS camera and a compact emission filter wheel. The performance of the proposed system was validated through experiments using reference dyes and a standard DNA amplification and detection. Furthermore, the system can be employed to detect four different fluorescence signals using other PCR chips that have a similar size with the PCB-based PCR chip presented. Further studies on the qPCR performance of the proposed system with different DNA concentration and actual clinical samples are required to determine the limit of detection and applicability.

When using industrial cameras for fluorescence detection, the samples are excited with LEDs for a set amount of exposure time. However, it is difficult to synchronize the exposure time and LED on time for open platform cameras. In the proposed system, the LEDs were fixed close to the reaction chamber, reducing the LED power required to compensate for the longer exposure time up to three-fold. Since open cameras are designed to fit smart phones, the lens size is significantly smaller than that used in industrial cameras in which the smallest lens (S mount) has a lens thread of 12 mm, enabling miniaturization of the overall fluorescence detection system. Taking into account that industrial cameras are not cost-friendly, it is evident that the open platform cameras hold great advantage when developing a portable and cost-effective device.

Although the photodiode sensor and the open platform camera are similar in cost owing to the development of smart phones, it is difficult to optimize the manufacture cost using a photodiode since an objective and/or ocular lens is required. The open platform camera has embedded optics that eliminates the need of external lens, also brining the advantage of ease of innovative development, assembly, and maintenance. For example, malfunctions such as reagent leakage or loading failures can be easily monitored since the reaction chamber can be observed simultaneously. In addition, this system holds great potential in delivering a more precise and accurate fluorescence data than the currently existing equipment using a different image processing algorithm and the fluorescence distribution profile within the reaction chamber. Given that open platform cameras with better performances are being developed along with the smart phone industry, the performance of the proposed system can be further improved.

## Figures and Tables

**Figure 1 sensors-21-06945-f001:**
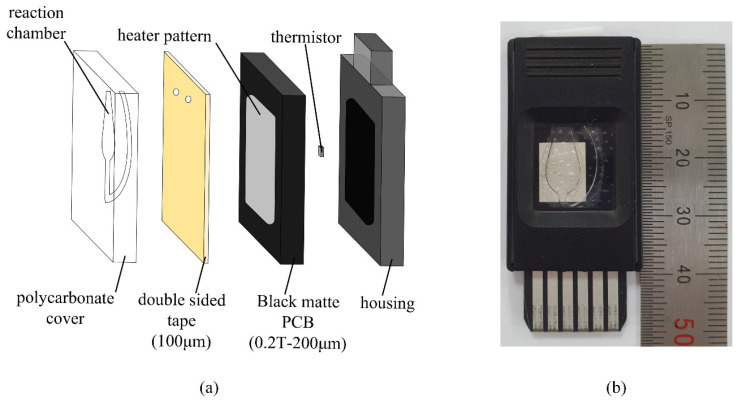
Schematic and actual PCR chip used in this study. (**a**) Schematic; (**b**) Photo of the actual PCR chip.

**Figure 2 sensors-21-06945-f002:**
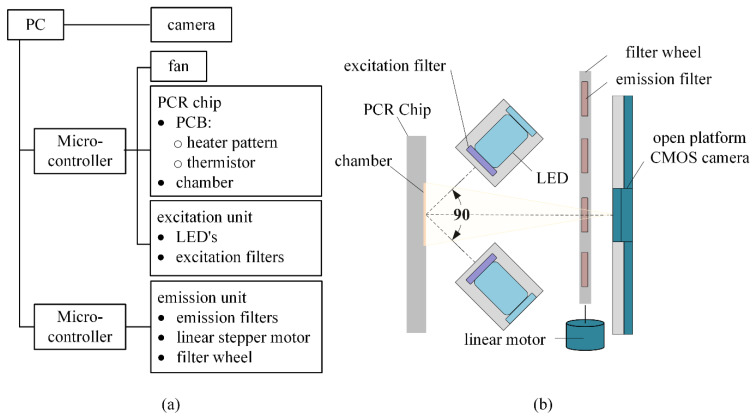
Functional block diagram and schematic of the proposed system. (**a**) Block diagram; (**b**) Schematic.

**Figure 3 sensors-21-06945-f003:**
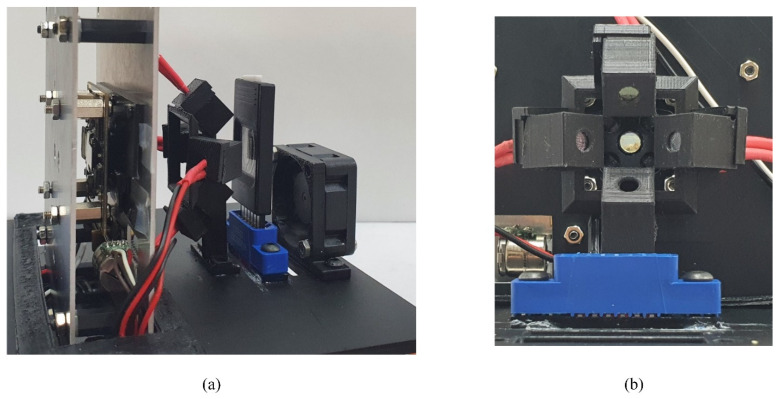
Overall four-color multiplex qPCR chip system and the excitation unit. (**a**) Overall four-color multiplexed qPCR chip system; (**b**) The front view of the excitation unit.

**Figure 4 sensors-21-06945-f004:**
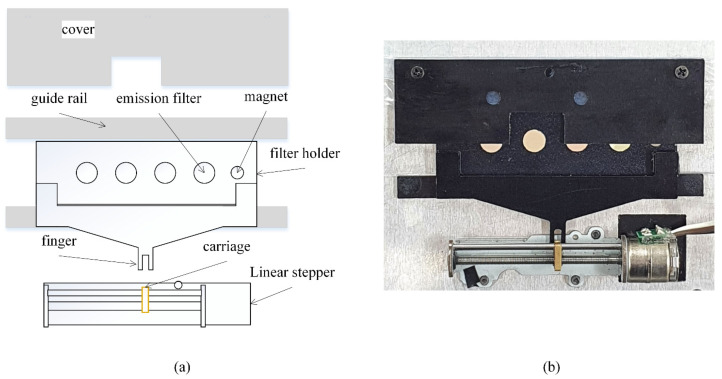
Emission unit. (**a**) Components of emission unit; (**b**) Assembled emission unit.

**Figure 5 sensors-21-06945-f005:**
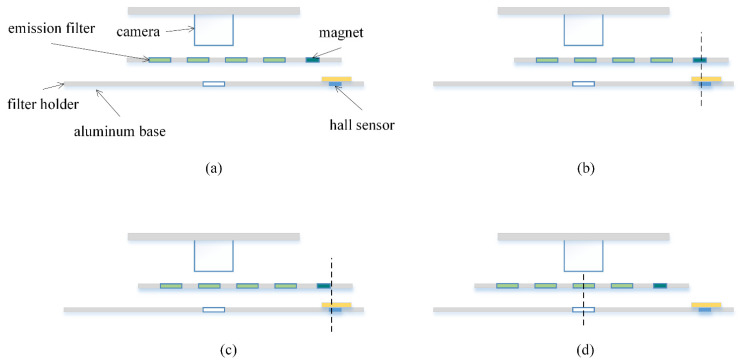
Filter wheel motion. (**a**) Schematic diagram of the filter wheel; (**b**) Coarse search for home position; (**c**) Fine search for home position; (**d**) Aligned filter position.

**Figure 6 sensors-21-06945-f006:**
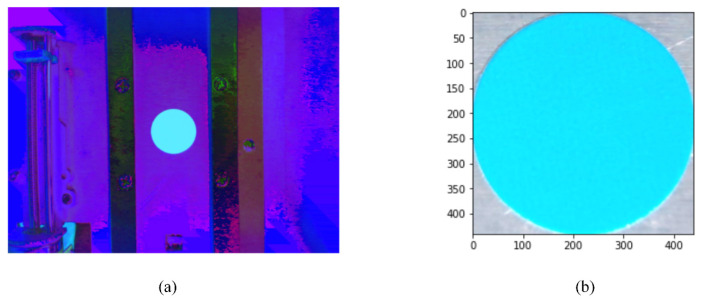
Images acquired in the motor precision testing fixture. (**a**) Image converted to HSV coordinate; (**b**) Image of the blue sticker in the bounding box.

**Figure 7 sensors-21-06945-f007:**
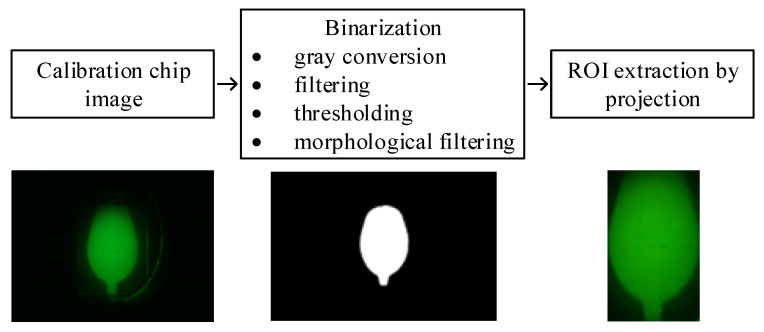
Image processing steps to obtain ROI, i.e., the chamber, for fluorescence analysis.

**Figure 8 sensors-21-06945-f008:**
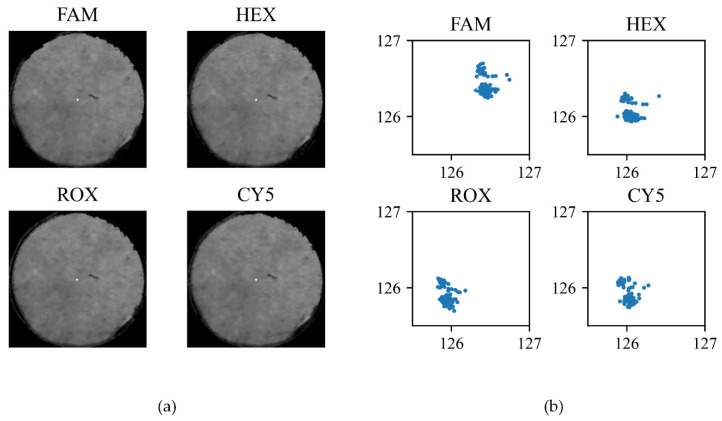
Variation of centroid location for each fluorescence image taken. (**a**) Images taken when each emission filter is aligned with the aluminum base hole. The centroid is represented as a white dot in the image; (**b**) Scatter plot of the centroids for 84 repeats.

**Figure 9 sensors-21-06945-f009:**
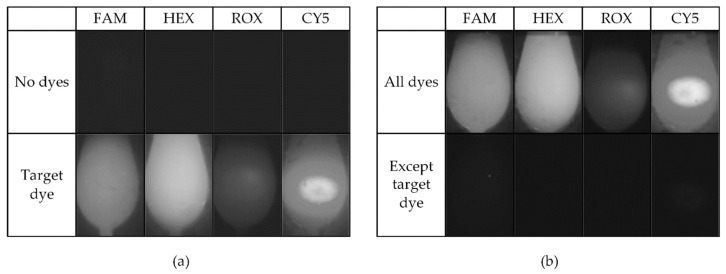
ROI images for individual dye experiments and for cross experimental results. These images have been gamma-collected by 0.5 for visibility. (**a**) Fluorescence detected in individual dye experiments; (**b**) Crosstalk experiments.

**Figure 10 sensors-21-06945-f010:**
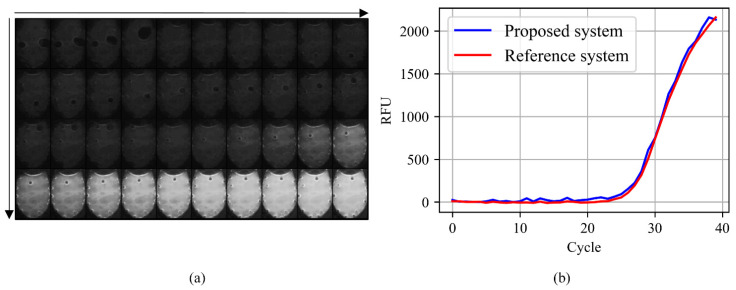
The change in the brightness of the chamber during 40 cycles (images have improved brightness for visibility), and the amplification curves from the proposed and reference system. (**a**) Fluorescence images of the chamber for 40 cycles; (**b**) Amplification curve of the reference system that uses a photodiode (red) and the proposed system with a CMOS camera (blue).

**Figure 11 sensors-21-06945-f011:**
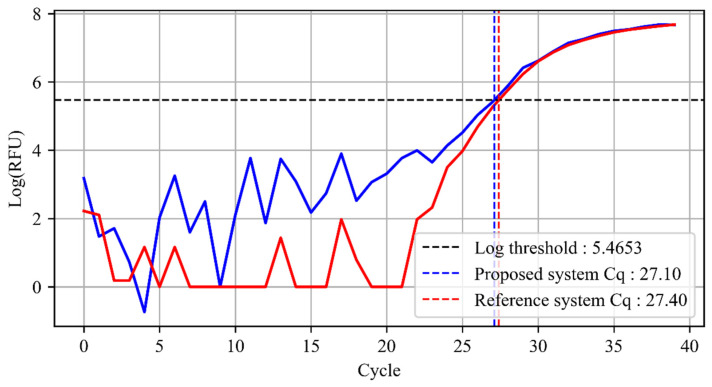
Cycle threshold obtained from the proposed system with a CMOS camera (blue) and reference system using a photodiode (red).

**Table 1 sensors-21-06945-t001:** Center wavelength (CWL) and full width at half-maximum (FWHM) of excitation and emission filters used in the proposed system.

Fluorescence	Excitation Filter		Emission Filter	
	CWL (nm)	FWHM (nm)	CWL (nm)	FWHM (nm)
FAM	470	30	520	20
HEX	530	20	565	22
ROX	570	20	615	40
CY5	630	20	665	20

**Table 2 sensors-21-06945-t002:** The part number, manufacturer, and millicandela rating of the LED mounted on the excitation device.

Fluorescence	Part Number	Manufacturer	Dominant Wavelength (nm)
FAM	C503B-BAS-CY0C0461	CreeLED, Inc.	470
HEX	C503B-GAN-CB0F0791	CreeLED, Inc.	527
ROX	LTL2P3KGKNN	LITEON	572
CY5	VLCS5830	Vishay Semiconductor Opto Division	624

**Table 3 sensors-21-06945-t003:** PCR sample composition.

Reagents	Concentration	Volume (µL)
DNA (*Chlamydia trachomatis*)	10^6^ copy	5.4 µL
Master mix	-	18 µL
Primer mix	0.28 pmole/µL	9 µL
DDW	-	3.6 µL
Total	-	36 µL

**Table 4 sensors-21-06945-t004:** Real-time PCR protocol.

Step	Temperature (°C)	Duration(s)	Cycles
Pre-incubation	50 °C	2 m	1
Pre-heating	95 °C	10 m	
Denaturation	95 °C	15 s	40
Annealing	60 °C	1 m	

**Table 5 sensors-21-06945-t005:** Result of individual dye and crosstalk verification experiments in ROI images of Figure 9.

Individual Dye Experiment					Fluorescence Crosstalk Experiment				
	Fluorescence					Fluorescence			
	FAM	HEX	ROX	CY5		FAM	HEX	ROX	CY5
No dyes	4.2	2.0	2.0	2.0	All dyes	75.8	96.7	22.5	80.5
Target	62.1	95.3	21.1	71.2	Except target	5.6	2.0	2.0	2.6
Gap	57.9	93.3	19.1	69.2	Gap	70.1	94.6	20.5	77.8
Relative gap	13.7	46.6	9.5	34.6	Relative gap	12.4	45.3	10.2	28.8

## Data Availability

Not applicable.

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
