# Peer review of "Cost-Effective Multiplex Fluorescence Detection System for PCR Chip†"

_sensors, 2021, doi:10.3390/s21216945_

Round 1

Reviewer 1 Report

The manuscript titled: "Cost-effective multiplex fluorescence detection system for PCR chip" by SH Yun et al. describes a low-cost fluorescence system for observing qPCR devices.  Overall the manuscript reads well and I believe would be of interest to many researchers aiming to create such platforms.  However, the following points must be addressed, most notably a quantitative number on what is meant by "low-cost", prior to publication.

P. 1, line 36, can some numbers on cost be added to articulate an acceptable or target price threshold?  

P.2, line 6, perhaps add a citation for billion-fold

P.2, line 26, what is meant by the capitalized "ASSURED"?  is it a standard, or ?

P.2, line 29, can the authors define cost-effective in USD?  For instance, a MyGO mini by Azure Genomics, a commercial multiplexed qPCR unit can be purchased for 7,000-8,000 USD now.
https://www.azuragenomics.com/mygo-mini-real-time-pcr-instrument.html

P.2., paragraph 30; why is CMOS better than PD/CCD?

P.4., Figure 2/description below are any collimation or focusing lenses required?  If so, please state.  How big is the camera pixel size?  Does the CMOS logic do any image processing prior to the data being conveyed?

P.5, para at line 4 and line 18, some details on the manufacture/model of the LEDs/filters would be useful.  ex. the OD, the cut-off wavelengths

P.7, how long does each FL measurement take at 4-channels due to the mechanical filter wheel movement?  How does this compare to the PCR cycle time?

P.8, line 24, list concentration in a more standard form

P.9, line 12, µL units are stated in various ways throughout manuscript.  

P.10/11, Figure 9 could be labelled more clearly; maybe DDW (no dyes) in 9a),  maybe All - Target in 9b bottom row; this took some time to understand.

P.11, line 1, sentence is unclear; gap and relative gap need to be better defined.  Is this a standard method of evaluation?

P.11, table 3, is this data from a single run, processed from figure 9, or does it represent multiple measurements per data point?  P.11, line 7, can be more clearly explained, and relates to this question.

P.12, it is easy to miss that the photodiode system is a commercial unit.  I would consider adding that to figures 10 and 11 in the legend to show comparable performance.  Maybe something like: "Commercial Unit, Photodiode"

P.12, figure 11, Does the data below 25 cycles matter if there is a disparity between your unit and the commercial unit?

P.13, line 8, is the first time exposure time was mentioned in the manuscript.  You should list the value used for the data acquisitions.  Was it 1s or 100s?

Figure 3 shows the unit, but is there a large cover over the entire system to prevent the influence of ambient light? 

Author Response

Dear Reviewer,

Thank you very much for the comments for the following paper:

Ms. Ref. No.: sensors-1397904

Manuscript Title: Cost-effective multiplex fluorescence detection system for PCR chip

The responses on your comments is in the attached file and start with ‘ >> ’ after each comments.

Kind regards,

Jong Dae Kim

Reviewer 2 Report

  • Please add scale on the photos in order to better visualize the chip size
  • p9 v12 "ul" font should be corrected

Author Response

(The authors gave the same response as above.)

Reviewer 3 Report

This referee considers that the proposed system is consistent and applicable as a method of analysis. They did an excelent work. However, it is necessary to comment on the advantages and disadvantages of this method compared to other similar products on the market. A table should be included showing the advantages and disadvantages of this system compared to others with similar characteristics. Otherwise it is not possible to have an idea, (from the point of view of the reader) about the novelty of the work.

Author Response

(The authors gave the same response as above.)

Round 2

Reviewer 1 Report

The authors have addressed my comments and suggestions.

Reviewer 3 Report

Ok